# Empowering People with a User-Friendly Wearable Platform for Unobtrusive Monitoring of Vital Physiological Parameters

**DOI:** 10.3390/s22145226

**Published:** 2022-07-13

**Authors:** Maria Krizea, John Gialelis, Grigoris Protopsaltis, Christos Mountzouris, Gerasimos Theodorou

**Affiliations:** 1Applied Electronics Laboratory, Department of Electrical and Computer Engineering, University of Patras, 26504 Patras, Greece; mkrizea@ece.upatras.gr (M.K.); g.protopsaltis@upnet.gr (G.P.); mountzou@ece.upatras.gr (C.M.); gtheodorou@upatras.gr (G.T.); 2Industrial Systems Institute, ATHENA RC, 26504 Patras, Greece

**Keywords:** wrist-wearable device, PPG processing, physiological parameters, web-based applications, data analysis

## Abstract

Elderly people feel vulnerable especially after they are dismissed from health care facilities and return home. The purpose of this work was to alleviate this sense of vulnerability and empower these people by giving them the opportunity to unobtrusively record their vital physiological parameters. Bearing in mind all the parameters involved, we developed a user-friendly wrist-wearable device combined with a web-based application, to adequately address this need. The proposed compilation obtains the photoplethysmogram (PPG) from the subject’s wrist and simultaneously extracts, in real time, the physiological parameters of heart rate (HR), blood oxygen saturation (SpO_2_) and respiratory rate (RR), based on algorithms embedded on the wearable device. The described process is conducted solely within the device, favoring the optimal use of the available resources. The aggregated data are transmitted via Wi-Fi to a cloud environment and stored in a database. A corresponding web-based application serves as a visualization and analytics tool, allowing the individuals to catch a glimpse of their physiological parameters on a screen and share their digital information with health professionals who can perform further processing and obtain valuable health information.

## 1. Introduction

Average life expectancy has increased over the years, resulting in a rise in senior populations [1]. The attitude of society towards senior citizens and their well-being is an indicator of its organization and civilization. Elderly people are a demographic that needs expert care and dedicated assistance, sometimes even on an everyday basis. They tend to feel even more vulnerable especially after experiencing health issues and having been released from a health care facility. This is a crucial point in their recovery and well-being, and they need all the help they can get, in either physical or virtual form. Assistive technology based on the Internet of Things (IoT) can support unobtrusive health monitoring at home with the use of electrical devices, such as sensors and other gadgets (wearable or not) that provide feedback and remote access to the end user, aiming at improving inhabitants’ quality of life by providing more independence and better care [2]. According to [3], existing smart home health monitoring technologies include physiological monitoring, functional monitoring/emergency detection and response, safety monitoring and assistance, security monitoring and assistance, social interaction monitoring and assistance, and cognitive and sensory assistance.

Treading on the groundwork of assistive technology, the aim of minimizing the hospitalization days of the elderly and sending them home without compromising their safety seems to be doable. The achievement of this goal has at least two advantages. First, the elderly benefit from returning home to a safe environment as soon as possible and this can work in favor of nurturing a positive psychology for them. Second, the health care system also profits from early yet safe discharge of the elderly to their home. The financial resources dedicated to the health sector are not unlimited [4] and the struggle to mitigate the consequences of the pandemic is ongoing. Unburdening the health care system is a tangible positive ramification of assistive technology development, and the work we envisaged could be a major abetment to this line of action.

Wrist-wearable devices are a common accessory of everyday life, worn by many people just to tell time in the beginning. With the integration of proper sensors, they evolved to non-invasive monitoring units that aggregate vital signals. With the application of befitting embedded algorithms, several physiological parameters critical for health status assessment can be extracted. These devices can also log activities, steps, calories and sleep patterns. The autonomy and portability of such an apparatus grants the user the possibility to wear it everywhere and at any time, which is a notable advantage. Identifying outliers or anomalies in heart rates and other features can help establish patterns that play a significant role in understanding the underlying cause of the demise of physical well-being. Additionally, the accumulation of this valuable information in a backend system, in a secure manner, can be leveraged to the advantage of the end users for optimizing their living standard. This can be accomplished by providing medical experts with the tools to interact with the data and draw valuable conclusions regarding health status and everyday lifestyle.

Light, the essence of photoplethysmography, known as PPG, is used to measure the volumetric variations of blood circulation. This measurement provides invaluable information regarding humans’ physiological parameters entailing their health status. PPG technology has been proven less complex operationally, more comfortable for the user and more cost efficient compared to other monitoring techniques [5]. The continuous detection and monitoring of human physiological parameters such as heart rate (HR), blood oxygen saturation (SpO_2_), and respiratory rate (RR) are of paramount importance. A wearable device that unobtrusively grants the elderly an opportunity to continuously and without any intervention extract those parameters constitutes a major achievement. The state-of-the-art techniques in modern practices approach the task of obtaining such measurements using validated pulse oximeters, which are worn on individuals’ fingers. These pulse oximeters are based on the transmission-mode PPG. The continuous monitoring of the human vital signals has shifted from simply an appealing idea to fitness enthusiasts into an everyday habit for a plethora of people using a smartwatch. Sensors integrated within smart watches use reflectance-mode PPG to gather vital signals. Although they are widespread amongst sportspeople, they have not been widely used in clinical practice [6,7]. This can be attributed to the fact that PPG signals are vulnerable to Motion Artifacts (MAs) caused by hand movements, which affect considerably the accuracy of the entailed physiological parameters [8,9].

PPG signal acquisition becomes increasingly challenging when additional factors such as environmental noise or sensor misplacement are considered, thus further affecting the accurate assessment of its features. Measuring physiological parameters utilizing wrist-wearable devices can be a more perplexing procedure compared with the finger or another part of the body due to the low blood perfusion in the wrist area. The design of a wrist-wearable device must consider factors such as the spacing of the light-emitting diodes (LEDs), photodiode surface, as well as biological factors such as skin tone. It is clear that efficient PPG processing must be performed to enable reliable reading of its features and consequently accurate extraction of the physiological values. All the aforementioned challenges highlight the need for the implementation of a device capable of monitoring physiological parameters continuously and in an unobtrusive manner, that can easily be operated without need for excess previous training.

This work is about presenting the architecture and the constituent components of a comprehensive user-friendly system which provides well-being monitoring services promoting peace of mind to senior citizens. The overall solution consists of two subsystems that are integrated via well-defined interfaces, but each one performs autonomous functions in an opaque manner: The subsystem of bio signal recording and physiological parameters extraction and the back-end subsystem of storage, data processing and services. The novelty of the proposed system predominantly lies on the minimal design of the wearable sensing device while entailing accurately all the parameters, allowing a least disturbing interaction with the end user. This system has the capacity to continuously extract in real time the physiological parameters of HR, SpO_2_ and RR, to perform well-being status assessment, and to provide personalized feedback to improve health status. Furthermore, in comparison with similar solutions, the proposed system has the advantage of functioning without the aid of a paired smartphone for collecting and delivering the bio parameters to the back-end system, thus enhancing its user-friendly attribute for users with low digital literacy.

The remainder of this paper is organized as follows: Section 2 introduces the main challenges regarding the monitoring of physiological parameters using wearable devices and gives details on related works and their limitations. Section 3 presents design and implementation aspects of the proposed integrated system at both physical and operational level as well as performance evaluation features of the proposed wrist-wearable device. Section 4 describes the implementation and the core components of the web-based application, which comprises a front-end to display physiological parameters to end-users, and a back-end to allow data processing and service provisioning. Finally, Section 5 concludes this paper with future research directions.

## 2. Related Work

As shown further, research has been conducted towards producing simultaneously more than one physiological parameter using the PPG method, through wrist-wearable devices as enablers of continuous health monitoring in everyday life. With the evolution of medical science and technology, the wearables are incorporated with multiple different sensors so that they can keep track of a wide range of measurements such as heart rate, blood oxygen level, body temperature and activity monitoring including many others. Nowadays, wearables play an important role in making health care practices more efficient and cost-effective. These devices can be connected to smart phones or web apps allowing people to store their data for future reference. An overview of such wearables follows.

Li et al. [10] predicted the outbreak of Lyme disease and inflammation by combining sensor data along with medical measurements. This work gave evidence of how wearables can monitor activity along with physiology.

Mishra et al. [11] detected COVID-19 by utilizing physiological (HR) and activity (steps) data acquired by wearable devices. 5200 subjects participated in the analysis, including individuals with COVID-19. The study indicated elevated resting heart rates relative to the subject’s baseline. Two algorithms—resting heart rate difference (RHR-diff) and heart rate over steps anomaly detection (HROS-AD)—were developed. The first algorithm was based on standardizing the resting heart rate over a fixed time frame to observe baseline residuals.

Seshadri et al. [12] performed a data-driven COVID-19 prediction employing an early detection algorithm (EDA) based on HR, HRV and RR collected from wearables devices. The EDA can detect physiological changes and alert users of possible infection with SARS-CoV-2 before they develop clinical symptoms.

Downey et al. [13] showed that only 16% of the subjects remained connected to obtrusive monitoring systems after 72 h. Furthermore, the cost for a complete vital sign monitoring system can be quite significant.

Zenko et al. [14] proposed a battery powered wearable device along with a simple algorithm for the extraction of the physiological parameters of HR, Pulse Rate Variability and SpO_2_. This work evaluates the acquired HR parameter while calibration and verification of the SpO_2_ parameter still needs to be performed.

Son et al. [15] introduced a wearable device which measures oxygen levels in the blood using a light reflection method while it integrates hardware for wireless data transmission. Experimental results were compared to the Texas Instruments development board (SpO_2_ AFE44x0 EVM), and the maximum deviation was 6.7% in HR measurement and 4.4% in SpO_2_ measurement.

Jarchi et al. [16] integrated the AC components of red and infrared PPG signals in a complex waveform and then by applying the bivariate empirical mode decomposition algorithm, the SpO_2_ value is estimated with an approximate error of 3%.

Preejith et al. [17] developed a wrist-based optical heart rate device which, in order to eliminate the noise, ignores the measurements when motion is detected. The accuracy of the HR measurements equals to 0.9, expressed as Pearson’s r. The coefficient r indicates the strength of a correlation between estimated and real values, and the magnitude of 0.9 nominates that the variables can be considered highly correlated.

Eugene et al. [18] designed a wearable device equipped with PPG sensors for extracting bio-information and the Centralized State Sensing (CSS) algorithm was developed for estimating HR. After comparisons on readings taken across sensors, it was proved that this specific algorithm achieved more accurate HR measurements.

Wojcikowski [19] proposed an algorithm for real-time HR estimation by a wrist-wearable device. The device incorporates PPG and accelerometer sensors. The acceleration signal is used to detect body movements which distort the PPG signal. The evaluation results evidenced that the developed algorithm for HR measurements outperformed the other algorithms from the literature.

Münzner et al. [20] presented methods for development of robust deep learning (DL) methods for human activity recognition (HAR) addressing the problems of normalization and fusion of multimodal sensor HAR data. The results show that sensor-specific normalization increases the prediction accuracy of the convolutional neural networks (CNN). In the context of multimodal HAR, further normalization techniques should be investigated which focus on other modalities such as physiological sensors.

Tang et al. [21] proposed a new CNN that uses hierarchical split (HS) for a large variety of HAR tasks, which can enhance multi-scale feature representation ability via capturing a wider range of receptive fields of human activities within one feature layer. Benchmarks demonstrated that the proposed HS module is an impressive alternative to baseline models with similar model complexity and can achieve higher recognition performance.

Zhang et al. [22] presented a Deep Neural Network (DNN) to detect lumbar-pelvic movements (LPMs), including flexion, lateral flexion, rotation, and extension, locally on-device, where the data were collected from a clinically validated sensor system. Continuous monitoring of these movements can provide real-time feedback to both patients and medical experts with the potential of identifying activities that may precipitate symptoms of low back pain (LBP) as well as improving therapy by providing a personalised approach.

Aside from the prototypes that emerged from literature research, as depicted in Table 1, there are wrist-worn commercial products available which utilize PPG sensors for obtaining physiological measurements, as shown in Table 2. Empatica E4 [23] is a wearable wireless multisensory device for real-time data acquisition and computerized biofeedback. E4 comprises four embedded sensing modules: a photoplethysmography (PPG) module, an electrodermal activity (EDA) module, a 3DOF accelerometer module, and a temperature sensing module. E4 offers the readings of HR, activity status and temperature while being capable of characterizing the function of the autonomic nervous system, EDA for assessing the sympathetic activation and HRV for assessing the parasympathetic activation. The device is compliant with international safety and emissions standards. MaxRefDes103# [24], is a physiological signal sensing band reference design available to the research community for further development. It is a wrist-worn wearable exhibiting high sensitivity and algorithmic processing capabilities comprising an enclosure and a biometric sensor hub with an embedded algorithm that processes PPG signals in real time for extracting HR and SpO_2_ only. Eventually, its corresponding output and raw data are streamed via Bluetooth to an Android application or PC GUI for demonstration, evaluation, and further elaboration. In addition to displaying the extracted HR and SpO_2_, the Android application furnishes additional algorithms for calculating RR, HRV, and sleep quality. Other wrist-worn wearables used for health monitoring are the Fitbit Versa 3 [25], Samsung Galaxy Watch 4 [26] and Apple Watch Series 7 [27]. The Fitbit smartwatch records the physiological features: HR, SpO_2_, skin temperature variation, sleep stages and RR measurement during sleep. Fitbit utilizes the BLE communication protocol and has mobile applications compatible with Android and iOS. Both the Samsung Galaxy Watch 4 and the Apple Watch Series 7 hold the capability for ECG and sleep monitoring. Moreover, both wearables calculate the HR and SpO_2_ physiological parameters and provide their data wirelessly through BLE/Wi-Fi and BLE correspondingly. The Samsung wearable is compatible with Android while the Apple wearable is compatible with iOS. As can be seen, among the approaches and the wearable solutions, there are limitations concerning the set of physiological values provided.

Our proposed comprehensive system introduces a non-invasive wrist-wearable device capable of capturing the PPG signal and extracting in real time the HR, RR, SpO_2_ physiological parameters simultaneously as well as rendering the raw PPG signal available which allows further processing for the sake of new bio parameters and features assessment. The extraction process takes place in situ ensuring the optimal utilization of its resources as well the network ones. It is a lightweight and embedded device with minimum add-ons exhibiting optimized memory capacity and processing power. It supports direct connection to 802.11.xx communication infrastructures which makes it an ideal candidate for instantaneous unhindered use in existing communication infrastructures offering high speed information sharing. Additionally, a cloud based back-end infrastructure offers all the required means to securely store the aggregated via https data in a time-series manner where end users and health professionals can perform visualization and algorithmic processing, respectively. 

## 3. Proposed Device Components

This section focuses on the design and development aspects of the proposed wrist-wearable device for the continuous monitoring of the PPG physiological signal and the extraction in real time of the HR, SpO_2_ and RR physiological parameters. Its scope is to provide an unobtrusive means to accurately assess the physiological data of the users’ enabling them to monitor their well-being status. The proposed design follows a modular approach both at physical (hardware modules) and at operational level (software modules) as described below.

### 3.1. Hardware Modules

The wrist-wearable device is a microcontroller-based device designed for continuous monitoring of PPG. The device extracts the HR, SpO_2_ and RR physiological parameters by implementing dedicated algorithms and transmitting the information over Wi-Fi to a developed web-based platform. The device is powered by a Lithium Polymer (LiPo) battery which can be charged using a USB cable. It is also equipped with an on-off switch for turning on or shutting down the device accordingly. Figure 1 depicts the block diagram of the proposed embedded device along with its external peripherals.

The hardware components of the device are surface mounted on a custom printed circuit board (PCB) which was designed considering effortless repair, analysis, and field modification of circuits with dimensions of 66.0 × 42.0 mm.

Microcontroller and RadioThe device’s microcontroller board incorporates the Espressif ESP-WROOM-02D, which is based on the ESP8266 chip implementing the Wi-Fi communication protocol [28]. Specifically, the core of the platform is the ESP8266 processor of Espressif systems, which is a Wi-Fi SoC integrating the full TCP/IP stack. The developed firmware code for the acquisition and the processing of the PPG signal, the extraction of physiological measurements and the code for the wireless transmission is being executed on the ESP8266 microprocessor. The ESP8266 integrates a Tensilica L106 32-bit RISC processor, achieving ultra-low power consumption and reaches a maximum clock speed of 160 MHz Moreover, ESP-WROOM-02D integrates an RF switch, matching balun, and a PCB antenna.Sensing ComponentsTwo sensing components are integrated into the embedded device, an optical sensor, and an accelerometer. MAX30102 is a high-sensitivity optical sensor able to continuously obtain the PPG signal and it is mounted on the standalone MAXREFDES117# reference board [29]. This low-power sensor board with a size of 12.7 mm × 12.7 mm is placed on the downside of the device’s PCB. In addition, the main board of PCB incorporates the Analog Device’s ADXL362 sensor, which is an ultra-low-power, 3-DOF MEMS accelerometer with measurement ranges of ±2 g, ±4 g, and ±8 g [30]. Power SupplyThe board is powered by either a Li-Ion 1600 mAh battery or the USB port. The power supply source is automatically switched on by the hardware. The battery is being recharged when a common USB power supply is connected to the USB port with a charging current set at 350 mA, controlled by the Microchip’s MCP73831 chip. The voltage of the power supply is stabilized by the Analog Device’s chip ADM7170 at 3.0 Volts and the board can harness all the available battery charge. The ADM7170 monitors constantly the voltage across the battery and cuts off the power supply of the circuit when the battery is not able to provide the correct voltage and current to the circuit.USB to Serial programmerThe USB port can also be used to program the ESP8266 without the need of an external programmer. Furthermore, it is possible to manually download the code using a switch and two buttons mounted on the PCB. The necessary translations of the USB protocol to Serial are performed by Silicon Labs, CP2102 chip.

A custom casing was designed to enclose the device using a 3D printer with PLA material. Figure 2 presents the printed circuit board of the device as well as the complete wearable device mounted on the wrist. The device is designed to be worn on the left hand and the placement is approximately 2 cm from the beginning of the wrist. Constant pressure between the PPG sensor and the skin is applied with the aid of the attached wrist strap. Inappropriate device placement results in insufficient light detected by the photodetector, a condition which activates a notification for proper alignment on the interface of the web-based platform.

### 3.2. Software Modules

The main operation of the proposed SW modules include

The adaptation of the optical sensor configurations to each individual user’s wrist and skin physiology;The processing of the accelerometer’s signal to detect the respective subject’s motionThe appropriate filtering of PPG signals;Their further processing for extracting HR, SpO_2_ and RR physiological parameters.

Figure 3 overviews the device’s algorithmic operation.

#### 3.2.1. PPG Acquisition

The principle of the device’s operation is PPG signal recording. During the initial stage, the Automatic Led Emission Control (ALEC) technique is performed, which algorithmically mimics the Automatic Gain Control closed-loop feedback circuit. ALEC automatically adjusts the system to the specific characteristics of each user, as it regulates the LEDs luminosity depending on skin tone and the diameter of the wrist of each user. During experimentation it was observed that the ADC output tends to alter in time. The former observation was attributed to the fact that even with perfect placement, the device is subject to slight movement, whereas the latter, to the unique physiology of each subject’s wrist and skin tone. To eliminate this problem, the LED level is adjusted until the ADC output reaches a satisfactory level that is optimal for the imminent signal processing and for not reaching saturation. This technique was developed to lead the system to a consistent response.

Subsequent to ALEC the device starts recording the IR and RED PPG signals along with acceleration data. The acceleration is monitored with the aim to detect conditions of intense movement. The segments of the PPG signal captured in circumstances of extensive movement impose catastrophic Motion Artifacts (MAs) onto the raw PPG signal, deeming the extraction of any physiological parameter unfeasible [31]. To overcome this, segments of the PPG where motion is detected are excluded and the sampling restarts. To detect motion, the built-in logic of the ADXL362 is utilized, whose activity and inactivity events are used as triggers for manipulating the PPG sampling. An activity event is triggered when acceleration of the device remains above a predetermined threshold for a specified time period. The accelerometer features two modes of operation, the absolute and referenced configuration. During the absolute type of operation, each incoming acceleration sample is compared with a user defined threshold, which when surpassed for a certain amount of time, signals that activity is detected. On the contrary, during the referenced mode of operation, each acceleration sample undergoes a regularization, to remove the effects of gravity, able to reach 1 g, and account for the status of the device prior to sampling. This is achieved by subtracting an internally determined value, captured regularly during inactive periods, from the acceleration sample. The corrected acceleration value is then compared with the user defined threshold, and in cases it is surpassed, an activity event is issued. Consequently, activity is detected only when the acceleration has deviated sufficiently from the initial orientation. The threshold selected for the activity event was set at 350 mg for at least 1 s. This helps to eliminate only movements whose intensity can obscure the physiological parameters estimation and allows lower intensity motion to be handled by the signal processing algorithm.

#### 3.2.2. Signal Processing

The PPG signal consists of AC and DC components. The DC component corresponds to non-pulsatile tissue, while the AC component alternates according to the heart cycle. Only the variable part of the signal is relevant for HR and RR determination; thus, the mean value is usually subtracted from the signal used in the HR and RR measurement. The recorded raw PPG signal is shown in Figure 4. 

The implemented device, which is thoroughly described in [32], deploys an algorithm for the digital processing of the PPG signal in the time domain to remove the effect of MAs and the DC component. Given the fact that the appropriate to our analysis frequencies of the PPG signal are ranging from 0.1 to 5.0 Hz [33], an IIR Butterworth bandpass filter with a passband of [0.1, 5] Hz is applied prior to signal manipulation. The variable component of the signal after filtering is shown in Figure 5.

Aiming at producing reliable estimations of the physiological parameters, extensive experiments were performed which resulted in the requirement for a PPG signal collection of at least 30 s. Within this time, useful raw PPG signal is certainly included, enabling the procedures of processing and physiological data extraction. Consequently, the measurements are produced every 30 s in cases of absence of considerable movement. As long as there are new measurements, the data are sent to the backend through Wi-Fi. Implementing the mentioned operating specifications, the device sustains a battery life of 48 h in continuous operation. 

#### 3.2.3. Heart Rate Estimation

For the estimation of the HR, the acquired signal of the infrared (IR) led source is utilized. After filtering the IR PPG signal, the Slope Sum Function (SSF) is applied [34], as shown in Equation (1). The procedure amplifies the peaks of each pulse and suppresses the noise represented by lower amplitudes.
(1)$$\mathrm{SSF}=\sum_{k=0}^{n}{\Delta x}_{k}{}^{2}{\;\mathrm{where}\;\Delta x}_{k}=\{{\Delta s}_{k}:{\Delta s}_{k}>0,\;0:{\Delta s}_{k}\leq 0\},$$

At this stage, the emphasized peaks of each window of the SSF output signal are identified as local maxima, as shown in Figure 6. After locating the peaks and estimating the time difference -d- between them, the instantaneous HR is computed for every pair of successive peaks using the Formula (2) [35]: (2)$${\mathrm{HR}}_{\mathrm{inst}}=\frac{6\times 10^{4}}{d},$$

Finally, the HR measurement is reckoned as the average of the instantaneous HR values in a 30 s time window.

#### 3.2.4. Blood Oxygen Saturation Estimation

To enable the assessment of the blood oxygen saturation in the blood, two LEDs, operating at the RED and IR wavelengths are utilized [36]. The principle of pulse oximetry is based on the comparison of the two waveforms, whose deviation is a direct indicator for the oxygen saturation. Their deviation occurs due to the different amount of light absorbed and emitted by the two types of hemoglobin, namely oxyhemoglobin and deoxyhemoglobin. Regarding the red wavelengths, deoxyhemoglobin absorbs a higher amount of light than oxyhemoglobin, while the opposite happens in the infrared region. Hence, the responses from the RED and IR LEDs captured from the photodetector are different.

The PPG waveform consists of two different components: the DC component corresponding to the light diffusion through tissues and non-pulsatile blood layers, and the AC (pulsatile) component due to the diffusion through the arterial blood. The developed algorithm locates the existing peaks and valleys and subsequently calculates the AC and DC components of both RED and IR PPG waveforms [37]. The DC component fluctuates slightly with respiration, while the AC component oscillates in concurrence with the changes appearing in the volume of arterial blood during the cardiac cycle [38]. Given the AC and DC components, a ratio R is calculated by the Equation (3):(3)$$R=\frac{{\mathrm{AC}}_{\mathrm{Red}}/{\mathrm{DC}}_{\mathrm{Red}}}{{\mathrm{AC}}_{\mathrm{IR}}/{\mathrm{DC}}_{\mathrm{IR}}},$$

Eventually, the SpO_2_ value is estimated using the Equation (4) provided by Maxim Integrated:(4)$${\mathrm{SpO}}_{2}=-45.006\times R^{2}+30.354\times R+94.84,$$

#### 3.2.5. Respiratory Rate Estimation

To perform RR estimation, two modulated signals need to be extracted from the original PPG signal obtained from the IR signal [39]. The two components, namely the Frequency Modulation (FM) and the Amplitude Modulation (AM), illustrate the effects of respiration as a physiological process on the PPG signal. Respiration is a complex process consisting of various mechanisms which cause many subtle changes to the original PPG signal. The most prominent of those effects regard FM which is the manifestation of the spontaneous increase in the heart rate during the inspiration and the corresponding decrease during expiration [40] and AM which is the result of reduced stroke volume during inhalation reducing the pulse’s amplitude [41].

Following the raw PPG acquisition, a bandpass filter is applied to eliminate frequencies not related to respiratory information. The process of peak characterization includes separating the waveform in individual pulses and detecting their maximum value. The FM can be defined as the time difference between two consecutive peaks as described in Equation (5), whereas the AM is formed by each individual amplitude peak of the signal as shown in Equation (6). Each time value assigned to the FM and AM signal samples is calculated as the mean of the time of occurrence of two peaks as shown in Equation (7).
(5)$$x_{\mathrm{FM}}=\left|t_{{\mathrm{peak}}_{i-1}}-t_{{\mathrm{peak}}_{i}}\right|,\;i=2,..,N,$$
(6)$$x_{\mathrm{AM}}=\left|y_{{\mathrm{peak}}_{i}}\right|,\;i=1,\;2,..,N,$$
(7)$$t=\frac{\left|t_{{\mathrm{peak}}_{i+1}+}t_{{\mathrm{peak}}_{i}}\right|}{2},\;i=2,..,N,$$

The values of the modulated signals are not homogenous, thus inhibiting the signal processing. To evenly sample the two waveforms, Shape-Preserving Piecewise Cubic Interpolation is performed on the acquired data points and the sampling rate is set at 4 Hz. Prior to the Fast Fourier Transform (FFT), a Hamming window is applied to minimize the side lobes of the frequency response. 

At this stage, the two power spectra are combined to amplify the potential peaks, the dominant frequency (F_d_) in the plausible range is identified and the final RR value is then computed by Equation (8):(8)$$\mathrm{RR}=F_{d}\times 60\;\left(\mathrm{breaths}/\min\right),$$

#### 3.2.6. Data Transmission

The wearable device transmits data to an API endpoint via Wi-Fi, implementing the HTTPS communication protocol which follows the HTTP protocol over a secure and encrypted connection. A time window of 30 s interposes between two data transmissions. A payload in URL encoded format is generated, which includes the values of physiological parameters collected by the wearable device; HR, SpO_2_, and RR. In addition, the payload includes the MAC address of the Wi-Fi’s Access Point (AP) and the battery level of the wearable device. The parameters of the payload are described in Table 3.

The API service, located on the server-side of the proposed web-based application, provides an endpoint to wearable device in which it can POST requests. Therefore, a parametric URL, presented in Figure 7, is utilized by the API service. By the time a POST request is applied to the API service, it parses the URL, deploys the GET variable to extract the value of each parameter, validates that each parameter has the appropriate format and stores the parameters into the database.

The MAC information is included as the proposed device has the capability of connecting to the Wi-Fi AP with the best signal strength in the case there is more than one Wi-Fi AP in the surrounding space. The AP with the highest signal strength will be the closest to the user. Thus, the MAC address of a particular AP can be exploited for assuming the approximate location of the user, given the location of each AP.

Typically, the SSID and the password of the Wi-Fi network are assigned into variables within the device’s code. This would require the end-users to enter their Wi-Fi credentials and upload new code on the device. To overcome this, the Wi-Fi Manager library is implemented, which allows end-users to connect to different APs without having to interact with the firmware. More specifically, when the device is activated by a user, a connection to a previously saved AP is attempted. If this process fails, the AP mode is enabled, allowing the user to configure a new set of SSID and password. The user has to navigate to a web page with default IP address 192.168.4.1 and enter his Wi-Fi credentials into a form. Once a new valid set of SSID and password is set, the device automatically reboots and establishes a connection.

The device also has the capacity to locally store data in cases of dropped or lost Wi-Fi connection. ESP8266 module provides the user with a flash memory of 1 MB, from which the 0.4 MB are occupied by the device’s firmware. The remaining available memory is utilized for storing the measurements produced during the absence of network connection. The saved data are transmitted when the Wi-Fi connection is restored and then the device continues its regular operation. The device extracts the physiological parameters every 30 s and transmits them along with other data. As mentioned, the 30 s time period is a specification, which emerged during the trials, since within this time period, useful raw PPG signal is definitely included, facilitating the production of reliable physiological data values.

#### 3.2.7. Measurements Evaluation

Aiming to evaluate the performance of the suggested wrist-wearable device and the accuracy of the corresponding extracted physiological parameters, commercial off-the-shelf certified devices were used. The values obtained by these devices are considered as reference and are compared with the values of the proposed wrist-wearable device.

Regarding the evaluation process of HR and SpO_2_ physiological parameters a medical finger pulse oximeter was utilized. The commercial finger pulse oximeter chosen is a certified medical device manufactured by Berry: BM2000D Bluetooth Pulse Oximeter [42]. The accuracy of RR determination methodology was evaluated utilizing the chest worn Zephyr BioHarness device, which is a physiological monitoring system with proven reliability in determining RR [43].

Ten healthy subjects with varying wrist circumferences and skin tones were provided with the wrist-wearable device and the reference devices. In particular, the subjects were equipped with the proposed wrist-wearable device and the Berry Pulse Oximeter along with the Zephyr BioHarness as ground truth devices. The experiments were performed at a sedentary state and the total duration of the experiment for each subject was 1 h, yielding an aggregation of 10 h of data.

To analyze the alignment between the data acquired from our proposed device and those from the reference instruments, the Bland–Altman plot was deemed ideal. The Bland–Altman graph consists of a plot of the difference between paired readings of two variables, in our case the derived and the reference values, over the average of these readings. Incorporated into the plot are the ±1.96 SD lines (the Confidence Interval) parallel to the mean difference line.

Figure 8, Figure 9 and Figure 10 present the comparative analysis of our data and render the proposed device a reliable system for the extraction of the desired physiological parameters.

The Bland–Altman plot displays four types of data misbehavior: systematic error (mean offset), proportional error (trend), inconsistent variability, and excessive or erratic variability. It should be noted that the Bland–Altman comparison method innately assumes that the two methods compared are portraying close results, a condition satisfied in our analysis. 

## 4. Web-Based Application

Nowadays, web-based applications dominate markets as they offer considerable advantages over traditional desktop applications. Web-based applications run directly on web browsers, demand minimal computer resources, do not require any installation process, are highly scalable, and maintenance tasks are performed centrally on the web server. In addition, they are portable and cross-platform available, which allows users to access them from any place and any type of device. Therefore, web-based applications are deemed to be the optimal solution for flexible projects. 

Web-based applications can be structured with various architectural patterns. The proposed one adopts the client-server model. The client-side refers to the visible and interactive part of the application, whereas the server-side is responsible for processing client requests and responses. In most cases, the server-side includes a database to support transactions with data and API services to support interoperability with end-devices and systems. 

### 4.1. Front-End Implementation

Front-end stands for the client-side of a web-based application. It provides a graphical interface, which allows end-users to interact with the web-based application and visualizes the information acquired by the wearable device. The front-end of the proposed web-based application employs the core web technologies; HTML5 structures the UI components, CSS styles the UI components, and JavaScript enables interactivity between the UI components and the end-users. In addition, a series of external frameworks and libraries were utilized to optimize the development process, i.e., to meet the business objective of the web-based application with minimal effort cost, achieve high performance standards, and ensure the appropriate infrastructure for further scalability. More specifically, Bootstrap framework, an open-source framework for front-end development, employed to accelerate the development process, optimize the overall performance in terms of computational resources, and provide a set of user-friendly and highly customizable UI components. This framework supports responsive design for web-based applications and incorporates web-accessibility standards. Chart.js, an open-source JavaScript library, employed for advanced chart implementation within the front-end. It offers pre-built and highly customizable UI chart components, allows efficient handling of data objects, optimizes the performance of charts’ drawing process, and improves the style of visualized data. jQuery, a lightweight and open-source JavaScript framework, employed to simplify event handling, improve the manipulation of UI components, and empower the interactivity capabilities.

The dashboard, shown in Figure 11, is the core interface of the front-end, and serves as personalized analytics overview and real-time monitoring tool. The primary objective of the proposed dashboard relates to the visualization of the user’s latest measurements, captured by the wearable device, in an intuitive and user-friendly way. Therefore, it employs UI components, mainly embedded line charts within cards, to depict the trend of each physiological parameter over the last one hour. Charts establish an asynchronous connection with the database; thus, they are automatically updated in a real-time manner with the latest measurements recorded by the wearable device. The horizontal axis of each chart represents the time, whereas the vertical axis represents the captured values of the specific physiological parameter displayed. Below each chart stands a card footer which informs end-users for the time of the chart’s data latest update.

Another core component of the front-end is the horizontal navigation menu bar, placed at the UI’s header area. It comprises a minimal information area, which displays the connection status of the wearable device and the battery level, as well as a list that maps the analytics and account areas of the web-based application. Analytics areas (Figure 12) visualize historical data for the corresponding physiological parameter in a user-friendly way. In addition, they implement a data picker filter to allow end-users apply the desired time span of the displayed data. 

Account area (Figure 13) displays a list with the available Wi-Fi networks and permit users to perform two (2) actions; edit network and delete network. The ‘edit network’ action allow users to change credentials of an existing Wi-Fi connection and ‘delete network’ action allows users to remove a specific network from the saved networks list. Moreover, from the ‘Account’ page, users can configure a connection with a new network.

### 4.2. Back-End Implementation

Back-end stands for the server side of a web-based application. On the back-end side, the web-based application implements embedded PHP scripts within the front-end code to provide dynamic functionalities. The dynamic aspect of the web-based application enables the capability to retrieve and visualize data from the database in a near real-time manner. The back-end comprises a RESTful API that allows submission of collected data from the wearable device into the database. More specifically, it parses a JSON object with the payload’s data and decodes it to extract each value and insert it into the database. A data validation process takes place to ensure the quality of data, such as checks for duplicate entries, appropriate data format and type, null values. In addition, SQL queries are executed to confirm that the extracted identifiers exist on the database to avoid conflicts on the data selection process.

A user authentication mechanism stands at the back-end, too. It has been implemented with Keycloak, an efficient, reliable, and extendable authentication and resource access management framework for web-based applications. The proposed application supports rights for two (2) roles, the administrator and the user. Administrators can access only an administrative dashboard page from which they can perform administrative tasks such as manual password resets, user account management, and service health monitoring. Users can access only the application UI resources—dashboard, analytics pages and account configuration page—from which they can keep up with visualized insights, apply changes related to their account information, and configure connections with networks.

The web-based application employs a MySQL database to perform transactions with data automatically collected by the wearable device and data manually provided by the end-users. Automatically collected data are included within the transmitted payload, i.e., physiological parameters, timestamp and MAC address. The database adopts a relational schema which structures the data into tables, as shown in Figure 14. The core functionality of the web-based application, i.e., data visualization, user authentication, and connection configuration, relies on four (4) tables; “Users”, “Devices”, “Connections” and “HealthRecords”.

The table of “Users” represents the entity of end-user. It stores information related to each end-user’s authentication credentials and the identifier of wearable device which is registered by the specific end-user. The primary key (PK) of this table is ‘UserId’ (int) which is automatically produced when a new end-user registers the web-based application. Therefore, auto-increment indexing has been declared to this attribute.The table of ‘Devices’ represents the entity of wearable device. It stores minimal information related to the matching between end-users and wearable devices. The primary key (PK) of this table is the attribute ‘DeviceId’ (varchar) which corresponds to the MAC address of each wearable device, as parsed from the transmitted payload.The table of ‘Connections’ stores information related to the Wi-Fi connection established between the wearable device and the local network. It stores information about the credentials of each connection, i.e., SSID and password. For each set of credentials, stands a ‘ConnectionId’ (int) which is automatically produced at the server-side with auto-increment indexing.The table of ‘HealthRecords’ stores information related to the physiological parameters. It has composite primary key (PK) which consists of the user identifier, wearable device identifier, and timestamp of the collected measurements. The attributes of user and wearable identifiers are foreign keys (FK) ‘Users’ and ‘Devices’, respectively. In addition, it includes attributes that represent all the physiological parameters collected by the wearable device, i.e., heart rate, SpO_2_, and respiratory rate.

## 5. Discussion

Physiological parameters provide critical information about individuals’ well-being status and signal early signs of a body dysfunction. For example, detecting an abnormally high HR could be an indicator that actionable measures should be taken to achieve healthy levels in order to reduce the risk of cardiovascular disease, while monitoring RR can detect early signs of a respiratory illness or allergy. SpO_2_ is useful in any setting where an individual’s oxygenation may be unstable or low for determining the sufficiency of oxygen or the need for supplemental oxygen.

Digital health technologies facilitate an individual-focused preventative approach through continuous monitoring of physiological parameters. This approach paves the way for personalized treatment, better care access and quality of service.

Providing that the subject be in a sedentary state, the proposed wearable apparatus introduced here, can unfailingly detect PPG signals and then reliably extract the physiological parameters of HR, SpO_2_ and RR. Specifically, the device achieves a mean percentage error equal to 2.47% and 0.8% for HR and SpO_2_, respectively, while estimating the RR parameter with a deviation of ±1.4 breaths per minute. The evaluation procedures showed that the wrist-wearable device can accurately detect fluctuations of the physiological parameters in a sedentary state. Consequently, it can be used to effectively monitor well-being status and provide valuable information. Moreover, a portable and cross-platform available web-based application has been developed, which serves as an informative and Wi-Fi connection establishment tool, from which individuals and health professionals can access the entailed parameters in a user-friendly and efficient way regardless the location, the type of device, and the operating system, and handle connections with networks in a simple way.

Impetus for future work is to enhance the accuracy of the extracted parameters in all respects. Our endeavor aspires to adopt an accelerometer-based detection and removal of faulty PPG segments or a signal preprocessing approach for MAs elimination. Alongside this, more trials with different subjects should be further performed. Collecting a larger amount of sample data from users of a broader spectrum in terms of age or skin color would also provide an opportunity for better algorithm calibration. Our future research interest is also focused on a multi wavelength photoplethysmography approach, which has proven superior performance than the single wavelength. Recently, advances on the sensing modality for detecting light from multiple sources enabled the development of a single chip sensor, removing the need for spectrometers, which have a prohibitive size for a wearable device. Moreover, in an effort to expand our knowledge on the effect of the light wavelength on the quality of the PPG signal, experiments are being conducted at wavelengths other than the red and infrared, which are currently used. Last but not least, such a device paves the way for injury prevention, early detection of illnesses or disorders, as well as early interventions with the aim to avoid the deterioration of health conditions.

## 6. Conclusions

This work presents in both physical and operational level all the discrete components of a comprehensive, user-embracive system able to unobtrusively record and process vital physiological parameters. It comprises a non-invasive wrist-wearable device able to incessantly detect PPG signals and solidly extract the physiological parameters of HR, SpO_2_ and RR and a multimodal web-based application via which the end users visualize real-time or historical data while allows health professionals to interact with that data for further algorithmic processing. The configuration of the system and its Wi-Fi connection was designed to be effortless even for older individuals that are not so much accustomed to high-end technology. The wrist-wearable device is a lightweight modular embedded device with a microcontroller based main board exhibiting optimized memory capacity and processing power, as well as long autonomy, and portable mounted off-the-shelf sensors for capturing the PPG signal. Moreover, by supporting direct connection to 802.11.xx communication protocols, it is an ideal device for the utilization of existing communication infrastructures that offer high speed information sharing. The cloud based back-end infrastructure offers all the required means to securely store the transmitted data from the wearable device over HTTPS protocol in a time-series manner. Both health professionals and end users themselves have easy access to historical and real-time data. The professionals can utilize the collected historical data to perform statistical analysis or execute AI/ML methods, aim to obtain valuable health information either for an individual or a group of them, thus unlocking a vast field of possibilities. The end users can glance at their data according to their preferences, simply by applying filters to adjust the friendly UI chart components of the front-end.

The accuracy assessment of the extracted physiological parameters, along with the evaluation of the system performance, were carried out against commercial off-the-shelf certified equipment, which was worn by healthy subjects with different anatomical characteristics.

Future actions include the increase in accuracy of the extracted parameters in all respects, the enhancement of the algorithmic processing capabilities and the execution of more trials on diverse subjects.

## Figures and Tables

**Figure 1 sensors-22-05226-f001:**
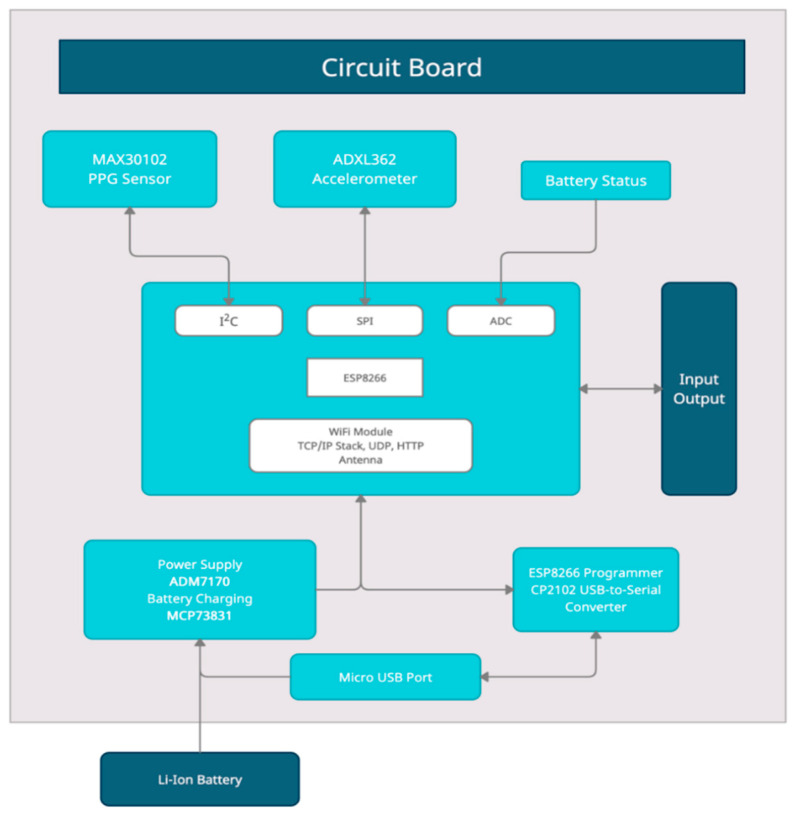
Block diagram of the developed wearable device.

**Figure 2 sensors-22-05226-f002:**
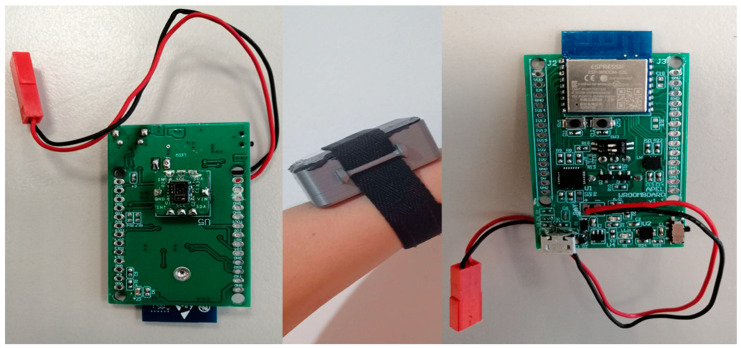
The proposed wearable device.

**Figure 3 sensors-22-05226-f003:**
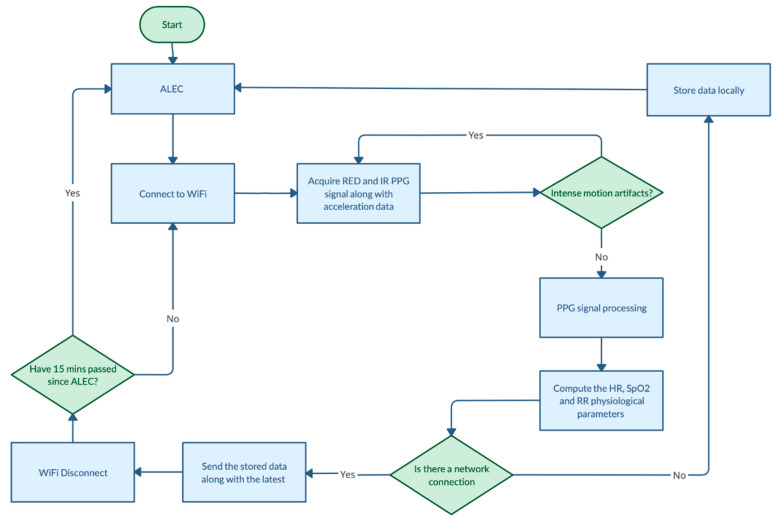
Flow chart of the algorithm.

**Figure 4 sensors-22-05226-f004:**
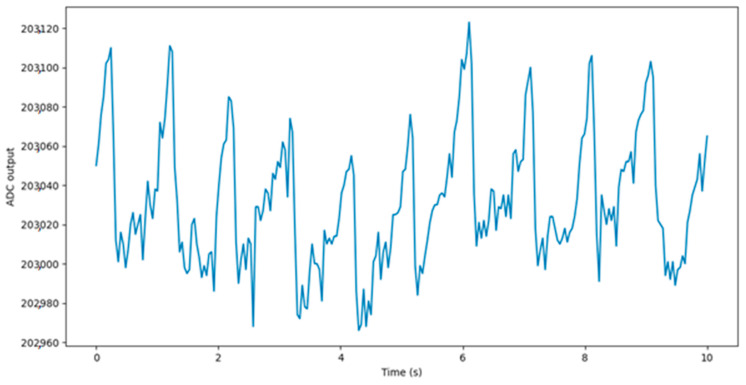
Raw PPG signal.

**Figure 5 sensors-22-05226-f005:**
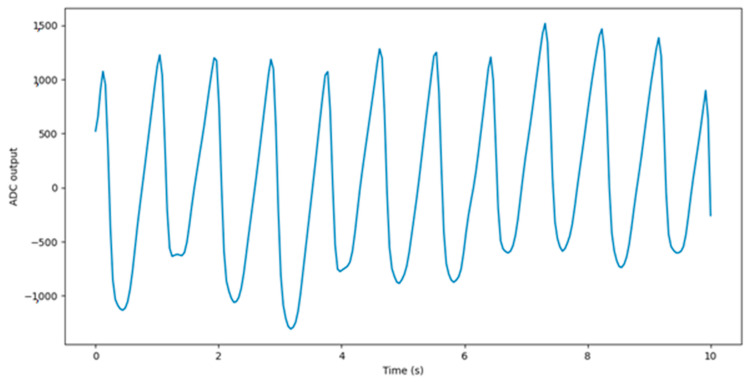
Filtered PPG signal.

**Figure 6 sensors-22-05226-f006:**
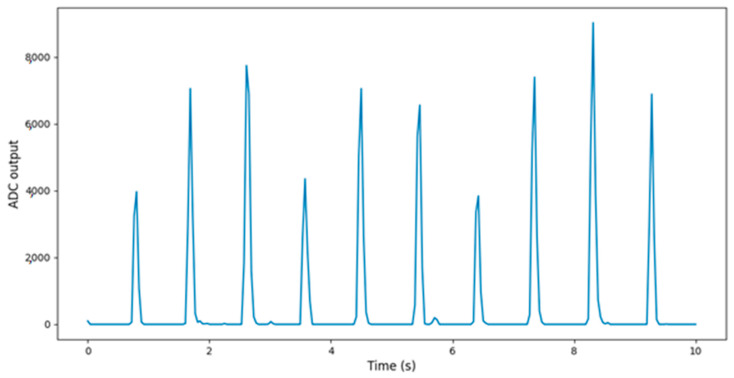
SSF output.

**Figure 7 sensors-22-05226-f007:**
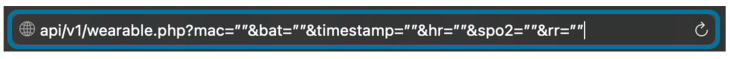
API service parametric URL.

**Figure 8 sensors-22-05226-f008:**
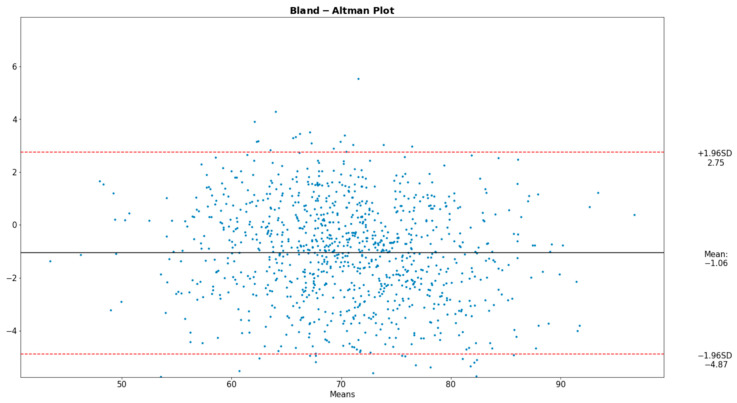
Bland–Altman plot for HR.

**Figure 9 sensors-22-05226-f009:**
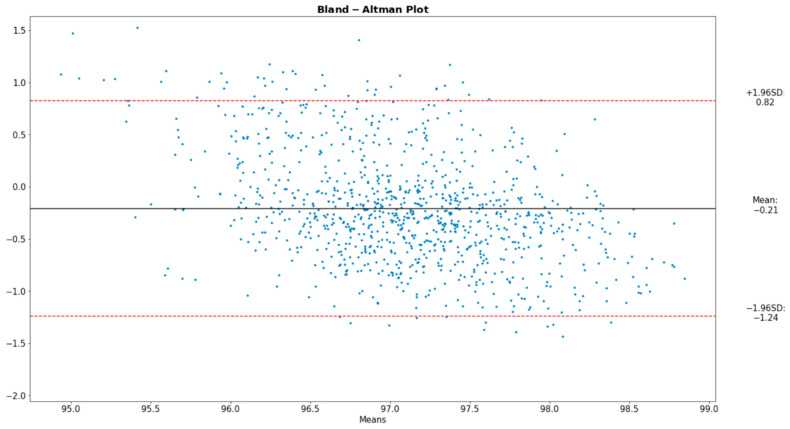
Bland–Altman plot for SpO_2_.

**Figure 10 sensors-22-05226-f010:**
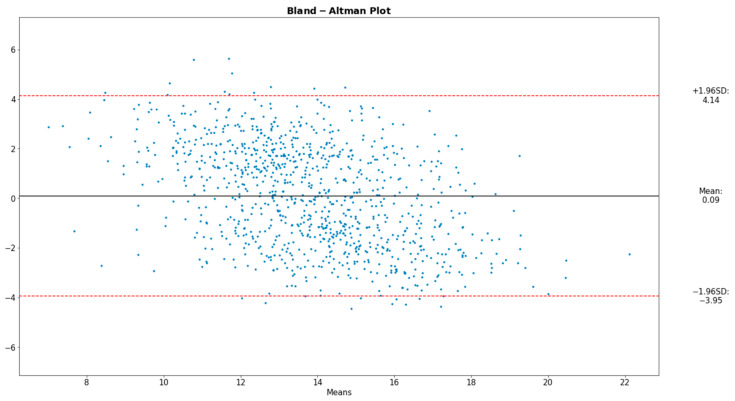
Bland–Altman plot for RR.

**Figure 11 sensors-22-05226-f011:**
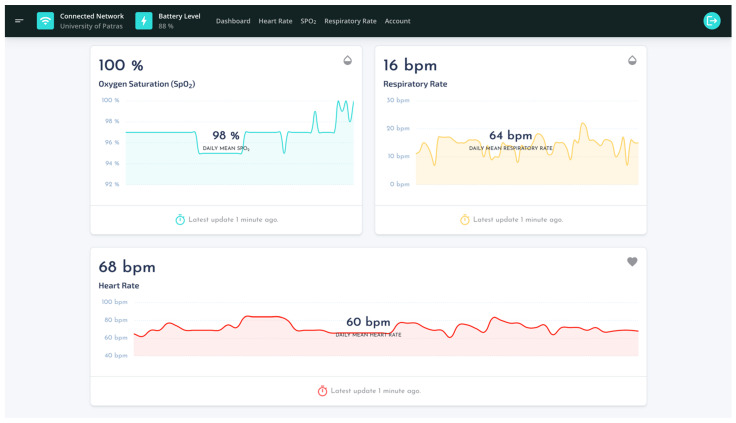
The UI of dashboard.

**Figure 12 sensors-22-05226-f012:**
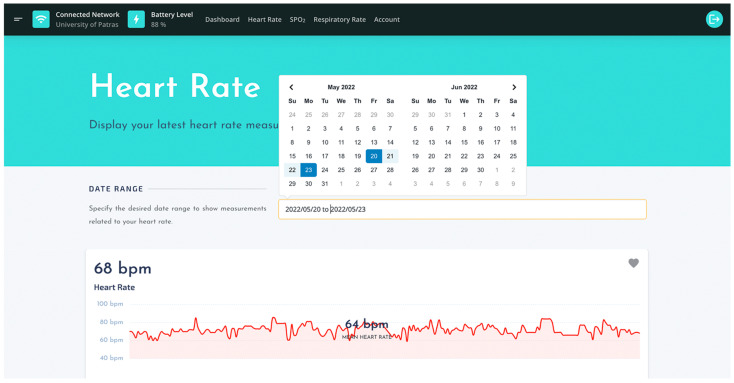
The UI of heart rate analytics.

**Figure 13 sensors-22-05226-f013:**
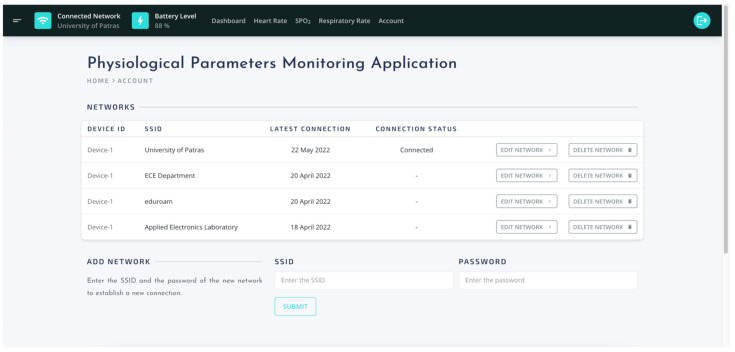
The UI of user account page.

**Figure 14 sensors-22-05226-f014:**
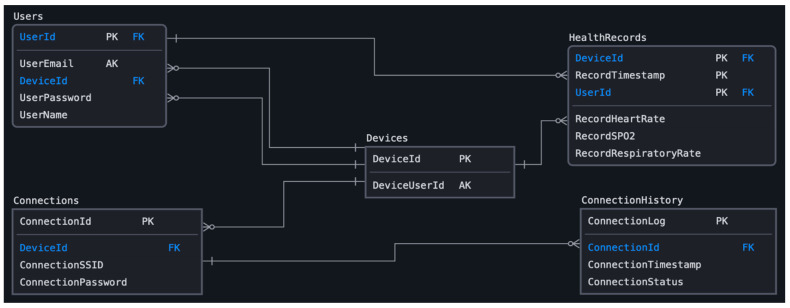
Database schema.

**Table 1 sensors-22-05226-t001:** Indicative functionalities and features of prototypes emerged from literature research.

Device Functionalities and Features	Proposed Device	Zenko et al. [14]	Son et al. [15]	Jarchi et al. [16]	Preejith et al. [17]	Eugene et al. [18]	Wojcikowski [19]
PPG	X	X	X	X	X	X	X
SPO_2_	X	X			X	X	X
HR	X	X	X	X			
RR	X						
Acceleration	X						
Continuous Measurement	X		X		X	X	
Communication Protocol	Wi-Fi	N/A ^1^	BLE	BLE	BLE	BLE	BLE
Battery life in streaming mode	48 h	NS ^2^	NS	NS	40 h	NS	NS
RT raw data	X	N/A	N/A	N/A	N/A	N/A	N/A

^1^ N/A: non-available. ^2^ NS: non-stated.

**Table 2 sensors-22-05226-t002:** Indicative functionalities and features of prototypes emerged from literature research.

Device Functionalities and Features	Proposed Device	Empatica E4 [23]	MaxRef Des103# [24]	Fitbit Versa 3 [25]	Samsung Galaxy Watch 4 [26]	Apple Watch Series 7 [27]
PPG	X	X	X	X	X	X
SPO_2_	X	X	X	X	X	X
HR	X	X	X	X	X	X
RR	X			X		
Acceleration	X	X		X	X	X
Continuous measurement	X	X	X	X	X	X
Communication protocol	Wi-Fi	BLE	BLE	BLE	BLE/Wi-Fi	BLE
Autonomy in streaming mode	48 h	24 + h	NS	12 + h	40 h	18 + h
RT raw data	X	N/A	X	Ν/A	Ν/A	Ν/A

**Table 3 sensors-22-05226-t003:** Parameters of payload.

Variable	Description
mac	MAC address of the Wi-Fi AP
bat	Battery level of wearable device
timestamp	The time when the measurements are extracted (in the form of Epoch Unix Timestamp)
hr	Heart rate
spo2	Blood oxygen saturation
rr	Respiratory rate

## Data Availability

Not applicable.

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
