# Peer review of "Empowering People with a User-Friendly Wearable Platform for Unobtrusive Monitoring of Vital Physiological Parameters"

_sensors, 2022, doi:10.3390/s22145226_

Round 1

Reviewer 1 Report

1) The paper states "The proposed compilation is of low-cost". Where is the evidence for this? Why does it it need to be low cost?

2) Most of the citations are incorrectly done, for example "Li et al. in their study [1] demonstrated" should be "Li et al. [1] demonstrated"; "as depicted in [4], " makes no sense, these errors exist throughout the document.

After reading the paper, sadly I cannot see anything new. Research already has made much proposals, such systems, indeed I am sorry to say better systems are reported to in the literature. Taking measurements and uploading to the cloud is not new.

Looking at the reference list, many many such wearable systems are missing.

Reviewer 2 Report

In this paper, the authors develop a user-friendly wearable device combined with a web-based application, which is used to obtain physiological parameters of Heart Rate (HR), Blood Oxygen Saturation (SpO2) and Respiratory Rate (RR). A detailed description is provided. The experimental results are corroborative and interesting. I can recommend it for a publication. There are still several concerns that need to be considered.

1. What are the main contributions in this paper? The authors should rephrase their main contributions or advantages in the introduction part.

2. The size of this wearable device looks still big. What is its main advantage compared with traditional smart watch?

3. The authors should explain how this manuscript advances this field of research and/or contributes something new to the research community in conclusion part.

4. In addition, I also recommend the authors to refer to several recent wearable based research literatures.

a. Tang, Y., Zhang, L., Min, F., & He, J. (2022). Multi-scale Deep Feature Learning for Human Activity Recognition Using Wearable Sensors. IEEE Transactions on Industrial Electronics, DOI: 10.1109/TIE.2022.3161812

b. S. Münzner, P. Schmidt, A. Reiss, M. Hanselmann, R. Stiefelhagen, and R. Dürichen, ''CNN-based sensor fusion techniques for multimodal human activity recognition,'' in Proc. ACM Int. Symp. Wearable Comput., Sep. 2017, pp. 158-165.

c. Zhang, Y., Haghighi, P.D., Burstein, F., Yao, L. and Cicuttini, F., 2021. On-Device Lumbar-Pelvic Movement Detection Using Dual-IMU: A DNN-Based Approach. IEEE Access, 9, pp.62241-62254.

Reviewer 3 Report

I would like to thank the authors for submitting this interesting work. Overall I feel that this is a good work and the technical contribution is novel and significant. However, I have some suggestions which the authors should incorporate in their manuscript.

1) The organization of the paper needs significant modification. For example, The introduction part lacks enough references to motivate the problem. For example, how smart home capabilities can be useful for elderly people and how their work is a vital step towards that direction. Some of the references that the author can find useful are:

[1] hapliyal, H., Nath, R. K., & Mohanty, S. P. (2017). Smart home environment for mild cognitive impairment population: Solutions to improve care and quality of life. IEEE Consumer Electronics Magazine, 7(1), 68-76.

[2] Maswadi, K., Ghani, N. B. A., & Hamid, S. B. (2020). Systematic literature review of smart home monitoring technologies based on IoT for the elderly. IEEE Access8, 92244-92261.

2) Line 43 and 44, "The characteristics of such an apparatus...." is unclear and needs explanation. If I understood it correctly, wrist worn devices cannot be used all the time due to battery limitations.

3) The information in the introduction from line 45 to line 63 looks like related work. Maybe the author can present the necessary information in a different way to make it look more like an introduction and motivate and introduce their problem.

4) The title of section 2 needs to be changed from background to related work. Further, from line 83 to 109, the author can consider reducing some information from this part. Moreover, I think it belongs to the introduction part where these information can be useful to motivate the problem and their proposed solution. This section can completely focus on the related work.

5) In line 157, author says E4 outputs HRV data, which I don't think is true. It does output the IBI data which is not same as HRV. Please check this fact.

6) In Table 1, E4 does output HR data. This needs to be checked. 

7) Line 191 and 192, this information is more suited either in the introduction section or in the conclusion section where the author can speak about the significance of their research.

8) Seems like there is a typo in line 202. Figure is mentioned twice.

9) In line 244, the author says the device is designed to be worn on the left hand. Why is this? I understand that such devices are best worn on the non-dominant hand. However, what about those who are left-handed. Can the device be adjusted so to wear in the right hand? This needs some clarification.

10) Line 369 to 375, these information is more suitable in the introduction part. Similarly for the other parameters such as Spo2 and HR. The author if wishes to talk about the background and relevance of these parameters should do so in the introduction part. This part should mainly focus on the method (what the authors did) with minimal reference to background.

11) In line 377, what is the frequency range for the bandpass filter used? Should be mentioned.

12) In line 390, I think the author should use a different notation for the dominant frequency as Fs is typically used for sampling frequency.

13) In Table 2, what is the unit of the extracted timestamp? should be mentioned.

14) Regarding figures 8, 9, and 10. I think the difference in the measurement between the proposed device and the reference can be better visualized with some more information using a bland-altman plot. The author should consider representing the results using a bland-altman plot by aggregating results from all the 10 users. 

15) The conclusion needs to be improved. It should highlight the significance of the research in relation to the bigger problem the author proposed in the introduction and what are the limitations and future work (if any) that needs to be done.

Round 2

Reviewer 1 Report

Comparing the current version to my last comments:

Last review comment "1) The paper states "The proposed compilation is of low-cost". Where is the evidence for this? Why does it need to be low cost?"

>>Sadly, the authors are not correct. They do not present a low cost system:

1. They ignore all the past research on how to design low cost systems.

2. The authors state "As far as the cost aspect is concerned, the prototype developed reached the amount of 302 approximately 50€ per device. Breaking down the cost per component the plus or minus 303 prices are 2€ for passive elements (L, R, C), 13€ for the ICs, 20€ for the sensing array, 3€ 304 for the various switches and leds, 10€ for the battery and 2€ for the PCB and cables". This makes no sense whatever. Mass production (which is the only way to do low cost) does not involve one-off component costs.

3. The paper states 50€ per device. The Apple watch costs less than that to make.

4. The authors ignore the costs for manufacture, reliability, aftermarket sales, software development, testing.......

 My advice is simple, just drop the low cost claims. In a paper where there is no research into low cost then it will be impossible to make any argument that the research is state of the art into low cost.

2) Most of the citations are incorrectly done, for example "Li et al. in their study [1] demonstrated" should be "Li et al. [1] demonstrated"; "as depicted in [4], " makes no sense, these errors exist throughout the document.

 >> The citations are all still wrong. For example "J. Zenko et al. [14] propose a" should be "Zenko et al. [14] proposed a"; "P. Son et al. [15] introduce a" should be "Son et al. [15] introduced a"; "Wojcikowski M. [19] presents an" should be "Wojcikowski [19] presented an". This error is throughout the paper, as pointed out before.

3) After reading the paper, sadly I cannot see anything new. Research already has made much proposals, such systems, indeed I am sorry to say better systems are reported to in the literature. Taking measurements and uploading to the cloud is not new.

>>The authors still do not present anything new. Dashboards are not new.

4) Looking at the reference list, many many such wearable systems are missing.

>>The authors add a few papers, but that’s it. Much of the state of the art literature is just missing as this is probably what the authors are not presenting novelty.

Looking at the new version:

1. The authors state "Currently, there is a lack of low-cost and reliable wrist-based wearable solutions for continuous monitoring of PPG" This is simply wrong.

2. The authors state "This system paves the way for creating a 103 safe space for the elderly to, even, live alone at their home, without feeling insecure. The lack of autonomy and self-sufficiency can be quite devastating for the emotional state of senior citizens and granting them the possibility to retain it with the help of unobtrusive wearable devices, is an excellent paradigm of the social aspect of the technological advancements and how they are positively affecting everyday life" 

>>This is all known, and indeed not worked on in the paper. 

3> Again, the authors do not state what their original research contribution is. It would be helpful for the authors to list their contribution claims like most researchers do.

Sadly, I am still at the opinion, as a researcher in this exact field for nearly 20 years, that this paper presents incorrect information (low cost), and does not present anything new.

Author Response

Response to Reviewer 1 Comments

 Point 1: The authors state "Currently, there is a lack of low-cost and reliable wrist-based wearable solutions for continuous monitoring of PPG" This is simply wrong.

Response: Thank you for your valuable comments. We have dropped the low-cost claims through out the text

 Point 2. The authors state "This system paves the way for creating a safe space for the elderly to, even, live alone at their home, without feeling insecure. The lack of autonomy and self-sufficiency can be quite devastating for the emotional state of senior citizens and granting them the possibility to retain it with the help of unobtrusive wearable devices, is an excellent paradigm of the social aspect of the technological advancements and how they are positively affecting everyday life". This is all known, and indeed not worked on in the paper.

Response: Thank you for your valuable comments. We have dropped the text and in general we are being focused on the capabilities of the system

 Point 3.  Again, the authors do not state what their original research contribution is. It would be helpful for the authors to list their contribution claims like most researchers do.

Response: Thank you for your comments. In the Introduction section, lines 99 – 113, we state our original research contribution.

 Point 4: Most of the citations are still incorrectly done.

Response: Thank you for the remark. All references have modified to be in line to your comment.

Point 5: Looking at the reference list, many such wearable systems are missing.

Response 4: Thank you for the remark. We have added three more references regarding high caliber commercial systems and extended Table1.  

Reviewer 3 Report

I thank the authors for submitting the revised version. All my comments are satisfactorily addressed. However, I have one minor comment regarding Figure 9 and 10. The texts in the figures are not readable. The figures can be improved.

Author Response

Response to Reviewer 3 Comments

Point 1: I thank the authors for submitting the revised version. All my comments are satisfactorily addressed. However, I have one minor comment regarding Figure 9 and 10. The texts in the figures are not readable. The figures can be improved.

Response 1: Thank you for your helpful remark. Figures are improved.
